A new sponge-associated starfish, Astrolirus patricki sp. nov. (Asteroidea: Brisingida: Brisingidae), from the northwestern Pacific seamounts

Zhang Ruiyan 1 2
Zhou Yadong 2
Xiao Ning 3 4
Wang Chunsheng wangsio@sio.org.cn 1 2 5
1 School of Oceanography, Shanghai Jiao Tong University , Shanghai , China
2 Key Laboratory of Marine Ecosystem Dynamics, Second Institute of Oceanography, Ministry of Natural Resources , Hangzhou , China
3 Laboratory of Marine Organism Taxonomy and Phylogeny, Institute of Oceanology, Chinese Academy of Sciences , Qingdao , China
4 Center for Ocean Mega-Science, Chinese Academy of Sciences , Qingdao , China
5 State Key Laboratory of Satellite Ocean Environment Dynamics, Second Institute of Oceanography, Ministry of Natural Resources , Hangzhou , China
Reimer James
Electronic publication date: 2020 May 27
Publication date: 2020
Volume: 8
Electronic Location ID: e9071
Received 2020 Jan 24; Accepted 2020 Apr 6
Copyright: ©2020 Zhang et al.
Copyright year: 2020
Copyright holder: Zhang et al.
License: This is an open access article distributed under the terms of the Creative Commons Attribution License, which permits unrestricted use, distribution, reproduction and adaptation in any medium and for any purpose provided that it is properly attributed. For attribution, the original author(s), title, publication source (PeerJ) and either DOI or URL of the article must be cited.
License URL: https://creativecommons.org/licenses/by/4.0/

Keywords: Starfish, Deep-sea, Brisingida, Astrolirus, Sponge associate, Seamount

Funding: Foundation of China Ocean Mineral Resources R & D Association DY135-E2-2-03 DY135-E2-2-06 Biological Resources Programme, Chinese Academy of Sciences KFJ-BRP-017-38 This work was supported by the Foundation of China Ocean Mineral Resources R & D Association (No. DY135-E2-2-03 and DY135-E2-2-06) and Biological Resources Programme, Chinese Academy of Sciences (KFJ-BRP-017-38). The funders had no role in study design, data collection and analysis, decision to publish, or preparation of the manuscript.

==============================
Seamounts are important deep ocean entities that serve as reservoirs for varied types of habitats and fauna. During the Chinese cruises in the northwestern Pacific seamount areas, a new starfish species of order Brisingida, Astrolirus patricki, was found at 1,458–2,125 m depth. All specimens of the new species were observed in situ attaching on hexactinellid sponges, suggesting a possible close relationship between the two taxa. A. patricki sp. nov. is the second known species of the genus, characterized by the abutting plates in the intercostal integument, separated first pair of adambulacral plates and densely distributed proximal spines. Phylogenetic analyses were conducted for order Brisingida to incorporate the new species as well as Hymenodiscus cf. fragilis (Fisher, 1906), Freyella cf. attenuata Sladen, 1889 and two Brisinga spp., for which we present the molecular data for the first time. Phylogenetic trees suggest a close relationship between A. patricki sp. nov. with Brisinga species rather than with Hymenodiscus species, which is inconsistent with morphological taxonomy. This study highlights the distinct morphological and ecological characters of the new species and provides new data for future investigation on Brisingida phylogeny.

Introduction

Seamounts are typical deep-sea biomes in the global ocean which harbor diverse types of habitats and benthic communities (Rogers, 2018; Victorero et al., 2018). Sponges are one of the dominant benthic fauna in seamount ecosystems, playing important ecological roles by providing habitat and settlement substrate for other seamount invertebrates, such as mollusks, hydrozoans and echinoderms (McClintock et al., 2005; Bell, 2008; Chu & Leys, 2010). Suspension-feeding brittle stars and crinoids with long and flexible arms are often observed perching on or wrapped around sponges (Hendler, 1984; Klitgaard, 1995; McClintock et al., 2005; Wulff, 2006). In this study, a new starfish species, Astrolirus patricki sp. nov., which was found attaching to deep-sea sponges, is reported based on five specimens from the northwestern Pacific seamounts.

Brisingidae species possess 7–20 spiny arms that are up to about 40 times the length of the disk radius (Downey, 1972; Downey, 1986). As exclusive deep-sea inhabitants, their long arms and spines potentially equip them to be excellent suspension feeders, stretching out and gathering food particles in the water column in the resource-diluted deep ocean (Emson & Young, 1994; Gale, Hamel & Mercier, 2013). Brisingidae is composed of 62 extant species designated in 10 genera (Mah, 2020). Genus Astrolirus Fisher, 1917 comprises only one species, Astrolirus panamensis (Ludwig, 1905), which was originally designated to genus Brisinga Asbjørnsen, 1856 and later proposed as the monotype in the new genus Astrolirus, differentiated from the other genera based on the presence of intercostal plates on arms and a pair of marginal plates between the first adambulacral plates (Ludwig, 1905; Fisher, 1917). This type species was discovered in the eastern Pacific Ocean at 1,820–2,418 m depth, with 1 eight-armed specimen and 27 nine-armed specimens of varying size (disc diameter 6–26 mm) reported (Ludwig, 1905). Thereafter, Astrolirus has seldomly been reported or investigated in the world ocean.

Astrolirus patricki sp. nov. herein described represents a new species in the genus. All five specimens of the new species are seven-armed and were captured from hexactinellid sponges (Fig. 1). Occasionally 2–3 individuals were spotted on the same sponge along with numbers of ophiuroids and crinoids. The new species differs greatly from A. panamensis in morphological characters and living habitat. In this study we present the taxonomic descriptions and illustrations of the new species, along with DNA barcoding sequences for each specimen. Phylogenetic tree for order Brisingida was constructed to incorporate the new species and new data from Hymenodiscus cf. fragilis (Fisher, 1906), Freyella cf. attenuata Sladen, 1889 and two Brisinga spp. on the basis of previous study (Zhang et al., 2019).

Figure 1 In situ photographs of Astrolirus patricki sp. nov.

(A) Holotype RSIOAS044. (B) Paratype RSIOAS028. (C) Paratype RSIOAS003. (D) Paratype RSIOAS052. (E) Paratype MBM286625.

Materials & Methods

Sample collection and examination

During the COMRA (China Ocean Mineral Resources R & D Association) cruises DY31, DY37, DY41, DY56 and a seamount cruise in the northwestern Pacific Ocean seamounts from 2013 to 2019, five specimens of the new species (Table 1) were collected by mechanical arms or siphon-pumps equipped on HOV and ROVs. Specimens were photographed in situ and on board by digital cameras. Tube feet tissues were extracted from each specimen and frozen in −80 °C refrigerator or liquid nitrogen for later molecular experiments, while other parts of specimens were preserved in 100% ethanol for morphological examinations. Morphological identification was conducted under a stereoscopic microscope (Zeiss Axio Zoom.V16). The type specimens of the new species are deposited in the Sample Repository of Second Institute of Oceanography (RSIO), Ministry of Natural Resources, Hangzhou, Zhejiang, China. Paratype MBM286625 is deposited in the Marine Biological Museum (MBM), Institute of Oceanology of the Chinese Academy of Sciences (IOCAS), Qingdao, China.

Table 1 Sampling information of Astrolirus patrickisp. nov. and four Brisingida specimens reported in this study.

Species	Specimen voucher	Cruise and station	Collection site	Depth (m)	Collection date	
Astrolirus patrickisp. nov.	holotype, RSIOAS044	DY41B, MCROV06	Weijia Seamount, northwestern Pacific Ocean, 156.41°E, 12.47°N	1935	2017.9.21	
	paratype, RSIOAS028	DY37-I, Dive105	Weijia Seamount, northwestern Pacific Ocean, 156.78°E, 12.96°N	1581	2016.4.30	
	paratype, RSIOAS003	DY31-III, Dive73	Caiqi Seamount, northwestern Pacific Ocean, 154.98°E, 15.22°N	1807	2013.9.9	
	paratype, RSIOAS052	DY56, ROV12	RD Seamount, northwestern Pacific Ocean, 149.85°E, 13.36°N	2125	2019.10.10	
	paratype, MBM286625	FX-Dive70	M2 seamount near the Mariana Trench, northwestern Pacific Ocean, 139.42°E, 11.27°N	1458	2016.3.27	
Brisinga sp.1	RSIOAS007	DY35-I, Dive83	Lamont Seamount, northwestern Pacific Ocean, 159.25°E, 21.61°N	1773	2014.7.29	
Brisinga sp.2	RSIOAS023	DY31-III, Dive70	Caiwei Seamount, northwestern Pacific Ocean, 155.55°E, 15.93°N	2431	2013.9.4	
Hymenodiscus cf. fragilis	RSIOAS009	DY37-I, Dive104	Weijia Seamount, northwestern Pacific Ocean, 156.51°E, 12.65°N	1957	2016.4.28	
Freyella cf. attenuata	RSIOAS037	DY38-III, Dive143	Mariana Trench, northwestern Pacific Ocean, 141.97°E, 11.82°N	4783	2017.5.23	

DNA extraction, sequencing and phylogenetic analysis

DNA extraction method, primer sequences and PCR programs were as in Zhang et al. (2017); Zhang et al. (2019). Five barcoding genes of each specimen, COI (cytochrome oxidase subregion I), 16S rDNA, 18S rDNA, 12S rDNA and H3, were acquired and the sequences were deposited in GenBank database (see Table S1 for accession numbers). K2P (Kimura 2-parameter, (Kimura, 1980)) pairwise genetic distances were calculated for COI genes in MEGA6 (Tamura et al., 2013).

For phylogenetic analysis, sequences of Brisingida were obtained from GenBank and accession numbers were shown in Table S1. Stichaster striatus and Cosmasterias lurida from the sibling forcipulatacean clade to the Brisingida (Mah & Foltz, 2011) were chosen as outgroup taxa. New data from A. patricki sp. nov. and four Brisingida species including Hymenodiscus cf. fragilis, Brisinga sp.1, Brisinga sp. 2 and Freyella cf. attenuata were added in the analysis on the basis of the previous study. The combining dataset includes 21 genospecies from 10 genera of the order Brisingida and 2 outgroup species. Phylogenetic analyses were conducted for the concatenated sequences of all 5 genes as well as for each independent gene. Models of evolution were estimated using jModelTest (Darriba et al., 2012) and the GTR + I + G model was selected as the best fit model. Maximum likelihood tree was constructed in raxmalGUI (Silvestro & Michalak, 2012) using GTR + I + G model with 1,000 bootstrap replicates. Bayes analysis was conducted in MrBayes (Huelsenbeck & Ronquist, 2001) using GTR + I + G model, running for 3,000,000 generations and sampled every 1,000 generation to estimate the posterior probabilities. The first 7,500 trees were discarded as burn-in. The tree topologies were observed and edited in Figtree v1.4.3.

Nomenclatural acts

The electronic version of this article in Portable Document Format (PDF) will represent a published work according to the International Commission on Zoological Nomenclature (ICZN), and hence the new names contained in the electronic version are effectively published under that Code from the electronic edition alone. This published work and the nomenclatural acts it contains have been registered in ZooBank, the online registration system for the ICZN. The ZooBank LSIDs (Life Science Identifiers) can be resolved and the associated information viewed through any standard web browser by appending the LSID to the prefix http://zoobank.org/. The LSID for this publication is: urn:lsid:zoobank.org:pub:7C8A768D-1312-498E-8352-BC4116D0B0F0. The online version of this work is archived and available from the following digital repositories: PeerJ, PubMed Central and CLOCKSS.

Results

Systematics

Order Brisingida Fisher, 1928	
Family Brisingidae G.O. Sars, 1875	
Genus AstrolirusFisher, 1917	

Diagnosis to Genus. Intercostal integument covered by thin plates; the first pair of adambulacral plates do not touch by their interradial faces, but are separated by a pair of marginal plates; first pair of marginal plates unit closely with a large interradial plate in the interradial faces.

Astrolirus patricki sp. nov.	
urn:lsid:zoobank.org:act:0F238379-CD10-4E51-A74A-0FEF055D72A5	

(Figs. 1–4)

Diagnosis. Arms 7, robust. Intercostal integument densely covered by irregular, abutting plates. No syzygy between proximal arm plates. The first pair of adambulacral plates separated by a pair of marginal plates. A large interradial plate above the first marginal plates, visible from the abactinal side, covered by scattered spinelets. Mouth spines and proximal adambulacral spines robust, densely distributed. Suboral spines 3–4; subambulacral spines 1–2, proximal ones truncate, capitate. One lateral spine to each adambulacral plate, starting from about the 8th. A pair of gonads to each arm.

Etymology. The name is originated from the character “Patrick Star” in the famous cartoon “SpongeBob Squarepants”, who always spends time with his best friend “SpongeBob”, a benthic sponge. Since all specimens of the new species were observed in situ living on sponges (Fig. 1), it was name by Patrick to reflect this curious relationship.

Material examined. Holotype, RSIOAS044 (Fig. 1A), 7 arms, of which 3 attached to the disk, others detached. r = 7.5 mm, the longest broken arm measures 153 mm without the missing tip. Length of genital area 35 mm, broadest part of arm measures 7 mm.

Paratype RSIOAS028 (Fig. 1B), 7 arms, including 3 regenerating ones, all attaching to the disk; r = 7 mm, the longest arm measures 153 mm, R/r = 21.9; length of genital area 36 mm, broadest part of arm measures nine mm wide. Paratype RSIOAS003 (Fig. 1C), 7 arms, all arm detached from the disk; r = 10 mm, R = 190–200 mm, R/r = 19–20; length of genital area about 50 mm, broadest part of arm 9–10 mm, at 15–20 mm from the disk. Paratype RSIOAS052 (Fig. 1D), 7 arms, r = 7.5 mm, the longest arm measures 168 mm, R/r = 22.4. Paratype MBM286625 (Fig. 1E), 7 arms, r = 7.5 mm, R = 171 mm, R/r = 22.8; length of genital area about 33 mm, broadest part of arm 7.5 mm.

Description of the holotype. Arms 7. Disk thick, elevated from the plane of arms (Figs. 2A–2C). Abactinal surface of disk covered by small rounded plates (Fig. 2G) bearing multiple sharp, thin spinelets, with few scattered pedicellariae between and at the base of the spinelets (Figs. 2C, 2F). Madreporite body locates at the margin of the disk, elliptical with a curved rift in the center (Fig. 2C). Papulae absent.

Figure 2 Astrolirus patricki sp. nov., abactinal view.

(A) Paratype RSIOAS028. (B) Paratype RSIOAS003. (C), (D), (H), holotype RSIOAS044, (C) Abactinal surface of disk and proximal part of arms, with red arrow pointing at the madreporite body, white arrow at the interradial plate and yellow arrows at the marginal plates. The red frame indicates the proximal region of arm connecting the disk and genital region, where pedicellariae do no form regular costae. (D) Abactinal surface of arm genital area with mosaic plating, red arrows show the costae bands. (E) Paratype RSIOAS003, abactinal surface of arm genital area, red arrows show the costae bands. (F) Paratype RSIOAS052, zoom in view of the abactinal disk, showing the multiple sharp spinelets on disk plates. (G) Paratype RSIOAS052, a piece of dissected skin from abactinal disk, shot from the inner side of the skin, showing the small round disk plates. (H) Abactinal surface at the middle of arm, black arrows indicate the pedicellariae bands.

Abactinal surface of arm within genital area densely covered by abutting plates of irregular scale forms. Pedicellariae scattered in the proximal region connecting disk and the genital area, not forming costae (Fig. 2C). Costae present within the genital area, about 30 in number; costae thin, densely located (two to each adambulacral plate), composed of raised band of pedicellariae (Figs. 2D–2E). Beyond the genital area, pedicellariae forms bands wider than costae, two to each adambulacrals (Fig. 2H). A pair of gonads to each arm. Each gonad with 3–6 oval pedals (Fig. 3G).

Figure 3 Astrolirus. patricki sp. nov. actinal view.

(A–D), (G), holotype RSIOAS044. (A) Actinal surface of the disk. (B) oral plates and spines; (C) Interradial angle between arms, red arrow shows the first marginal plates, yellow arrow shows the second marginal plate. (D) Lateral view of the disk, showing the conjunction of plates in the interradii. Red arrows show the first marginal plates, yellow arrows show the first adambulacral plates, white arrow shoes the interradial plate. (E) Paratype RSIOAS003, lateral view of the disk, red arrows show the first marginal plates, yellow arrows show the first adambulacral plates, white arrow shoes the interradial plate. (F) Paratype RSIOAS052, adambulacral plates and spines at the middle of arm, yellow arrows show the subambulacral spines, white arrows show the furrow spines, red arrows show the lateral spines. (G) One of the paired gonads and digestive caeca in genital area.

Figure 4 Astrolirus. patricki sp. nov. holotype RSIOAS044, mosaic image of abactinal arm.

Yellow arrows show the subambulacral spines, white arrow shows the aboral furrow spine, black arrow shows the adoral furrow spine, red arrows show the lateral spines.

Adambulacral plate in proximal area subquadrate, elongated in middle and distal part of the arms (Fig. 4). Proximally the ambulacral groove almost completely concealed by adambulacral spines (Fig. 4). The first pair of adambulacral plates on adjacent arms entirely separated by a pair of marginal plates (Figs. 3C–3D). No syzygy between proximal plates. The first pair of marginal plates unit closely with an interradial plate (Fig. 3D). Interradial plate large, nearly naked, extending to the abactinal side of disk (Fig. 2C). Proximally 2–3 marginal plates well-developed, present on the lateral side of the arm (Fig. 2C). The following marginal plates degrade to a line of protuberances on the side and corresponding to each adambulacral plates, bearing long lateral spine. Lateral spine present from about the 8th adambulacral plate, one to each plate (Figs. 3F, 4). The longest measures 15–20 mm.

Adambulacral plate armature includes: (1) 2 subambulacral spines and 1 aboral furrow spine, forming a diagonal row; the outer subambulacral spine usually the largest and most robust; (2) 1 adoral furrow spine, thinner than the aboral one (Fig. 4). The first pair of adambulacral plate bear a diagonal line of 4–5 spines, two on the surface of the plate, and 2–3 smaller ones on the distal furrow corner (Figs. 3B–3D). At middle and distal part of arm, the number of subambulacral spine on each adambulacral plate reduces to 1 (Fig. 4). Proximal subambulacral spines truncate form, capitate (Fig. 3C). The furrow spines become shorter and thinner on distal plates. All adambulacral spines bear pedicellariae, leaving only the tip of the spines naked.

Mouth plate large, bearing 3–4 robust suboral spines, forming a diagonal line, similar in form with the proximal subambulacral spines (Figs. 3A–3B); 2 small furrow spines present on the aboral corner of the plate, close to the first adambulacral plate; on the oral margin, the plate bears 2 very minute spines, one pointing to the actinostome and one pointing rather to the furrow, forming a sharp angle with the former. All oral spines bear pedicellariae.

Variations in paratypes. In the paratypes, adambulacral plates at middle and distal part of arm bear equal number of spines as the proximal adambulacral plates (Fig. 3F), instead of having one less subambulacral spine as in the holotype. In several undeveloped arms in the paratypes, gonads were not spotted. The shape and size of the marginal and interradial plates vary slightly in paratype RSIOAS003 (Fig. 3E).

Coloration. Color in life orange (Figs. 1, 2A–2B).

Distribution. Known from the northwestern Pacific Ocean, on seamounts, 1458–2125 m depth.

Molecular and phylogenetic results. COI K2P distances between specimens of A. patricki sp. nov. are less than 0.003, which are considered to be intraspecific distances. A. patricki sp. nov. is closest to Brisinga species, with distances between 0.123–0.159. The topology of the Bayes tree is shown in Fig. 5 and the nodes marked by black dots are support by both Bayes tree and ML tree. This tree is overall in line with results of the previous studies (Mah & Foltz, 2011; Zhang et al., 2019).

Figure 5 Phylogenetic tree of order Brisingida including Astrolirus patricki sp. nov. and 4 new specimens based on a concatenated dataset of COI, 16S, H3, 12S and 18S genes.

Topology follows the result of Bayes tree, bootstrap values and posterior probabilities are shown for each node. Nodes marked by black dots are support by both Maximum Likelihood Tree and Bayes Tree. The new species and new data reported in this study are colored red in the tree.

Remarks for A. patricki sp. nov. Types of A. patricki sp. nov. from different seamounts show little intraspecific divergences other than size and adambulacral spine number. The difference between A. patricki sp. nov. and the type species of the genus, A. panamensis, are listed in Table 2. Although both species are characterized by the presence of abactinal plates, in A. panamensis the plates locate between the transverse costae, whereas in A. patricki sp. nov. a mosaic of abutting plates forms the abactinal arm skeleton. The two species further differ in arm number, adambulacral spine number and distribution, as well as the number of genital organs. It should be noticed that, in Ludwig’s original description of A. panamensis, “several, consecutive pairs of branched gonads on each arm (4–5 in one male specimen)” were recorded, each branch has their own gonopore (Ludwig, 1905), but later when Fisher established the new genus, he described Astrolirus as having 2–4 gonads to each arm (Fisher, 1917). Whether this new description resulted from a reexamination of the type specimens was not clearly indicated (Fisher, 1917). Therefore, in this study we choose to follow Ludwig’s original descriptions when evaluate the differences between the new species and A. panamensis.

Table 2 Major morphological differences between A.patrickisp. nov. andA. panamansis.

Diagnostic characteristics	A.patrickisp. nov.	A. panamansis	
arm number	7	8–9	
costae	raised bands of pedicellariae	a transverse line of up to 11 strong, thick plates	
intercostal plates	abutting	usually isolated by small distances	
adambulacral plate armature	1–2 subambulacral spines, 1 adoral and 1 aboral furrow spines	1 subambulacral spine, 1 adoral and 1 aboral furrow spines in most specimen	
lateral spines	1 to each adambulacral plate	on every second or third adambulacral plate (in irregular change)	
suboral spine	3–4	1	
genital organ	1 pair to each arm	several, consecutive pairs on each arm (4–5 in one male specimen)	

In addition, A. patricki sp. nov. is characterized by a large number of suboral spines (3–4), whereas in A. panamensis, only 1 suboral spine present on each oral plate. Interestingly, in A. panamensis, some large specimens (r = 13 or 10.5 mm) have the 2 suboral spines on a pair of oral plates covered in a same membrane (Ludwig, 1905). This characteristic has only been reported in Brisinga andamanica Wood-Mason & Alcock, 1891 and Brisinga bengalensis Wood-Mason & Alcock, 1891 (Alcock, 1893), whose biological function is not clear.

Discussion

The COI genetic distances (<0.003) fall in the range of intraspecific distance suggested by Ward, Holmes & O’Hara (2008) for Asteroidea, which confirmed that specimens of A. patricki sp. nov. are indeed the same species. Based on the current dataset, the new species is genetically closer to Brisinga species (0.123–0.159) rather than to Hymenodiscus Perrier, 1884 species (0.194–0.203). The former distance barely falls out of the range of congeneric divergence of Asteroidea, whereas the latter is an undoubted intergeneric distance (Ward, Holmes & O’Hara, 2008). Similar results were also suggested by the phylogenetic analyses. In the Brisingidae clade, A. patricki sp. nov. appears to be intermediate between the Brisinga clade and Hymenodiscus spp., forming a sister clade with Brisinga spp. and Freyellaster fecundus (Fig. 5). The molecular evidence is to some degree inconsistent with the morphological taxonomy. Based on morphological characteristics, Hymenodiscus and Astrolirus could be differentiated from other Brisingidae genera in having the paired marginal plates between the first adambulacral plates. Hymenodiscus further differs from Astrolirus in having bare integument between costae. Previous cladistic analysis on Brisingida phylogeny also suggest that the two genera should cluster as a derived clade (Mah, 1998a; Mah, 1998b). The reason for such a discrepancy between molecular and morphological evidence is not clear yet. In addition, several nodes in the ML tree were support with low bootstrap value (Fig. 5), possibly owing to data deficiency. Genetic data from other Brisingidae species and genera should be supplemented in the future for a comprehensive phylogenetic analysis.

Conclusions

The new species Astrolirus patricki sp. nov. reported in this study represents the second known species in the genus, which is distinguished from its congener by the form and organization of costae and intercostal plates as well as numbers of spines and arms. Morphological description and molecular data delimitate the new species and provide reference for future taxonomic and phylogenetic study of related species. The current phylogenetic analysis on order Brisingida indicates an intermediate position of A. patricki sp. nov. between Brisinga and Hymenodiscus, but more samples and multi-gene analysis are needed in the future to clarify the actual systematic and phylogenetic relationships among these genera.

Supplemental Information

Table S1 GenBank accession numbers of taxa used in phylogenetic analyses

Click here for additional data file.

Supplemental Information 1 DNA sequences

Click here for additional data file.

We thank all the scientists and crew on the R/V XIANGYANGHONG 9, R/V HAIYANGLIUHAO, R/V DAYANGYIHAO and R/V KEXUE for their work in the collection of the specimens. Dr. Christopher Mah from Smithsonian Institution and Dr. Anna Dilman from P. P. Shirshov Institute of Oceanology are thanked for their suggestions on the examination of the new species. We also thank Dr. Dongsheng Zhang, Dr. Chengcheng Shen, Dr. Dong Sun, Dr. Bo Lu and Jieying Na in our lab for their helps in sampling, experiments and suggestion on the writing of this article.

Additional Information and Declarations

Competing Interests

Author Contributions

Field Study Permissions

DNA Deposition

Data Availability

New Species Registration

The authors declare there are no competing interests.

Ruiyan Zhang conceived and designed the experiments, performed the experiments, analyzed the data, prepared figures and/or tables, authored or reviewed drafts of the paper, and approved the final draft.

Yadong Zhou performed the experiments, prepared figures and/or tables, and approved the final draft.

Ning Xiao analyzed the data, prepared figures and/or tables, and approved the final draft.

Chunsheng Wang conceived and designed the experiments, authored or reviewed drafts of the paper, and approved the final draft.

The following information was supplied relating to field study approvals (i.e., approving body and any reference numbers):

Field experiments were approved by the China Ocean Mineral Resources R & D Association (DY31, DY37, DY41, DY56).

The following information was supplied regarding the deposition of DNA sequences:

The sequences are available at GenBank: MN879468 to MN879485, MN879536 to MN879543, MN885899 to MN885907, MN963767 to MN963772, MT127565.

The following information was supplied regarding data availability:

The holotype specimen RSIOAS044 and the paratype specimens RSIOAS028, RSIOAS003, RSIOAS052 are deposited in the Sample Repository of Second Institute of Oceanography (RSIO), Ministry of Natural Resources, Hangzhou, Zhejiang, China. Paratype MBM286625 is deposited in the Marine Biological Museum (MBM), Institute of Oceanology of the Chinese Academy of Sciences (IOCAS), Qingdao, China. GenBank accession numbers are available in Table S1.

The following information was supplied regarding the registration of a newly described species:

Publication LSID: urn:lsid:zoobank.org:pub:7C8A768D-1312-498E-8352-BC4116D0B0F0

Astrolirus LSID: urn:lsid:zoobank.org:act:ACD4F6B2-1246-4A16-8861-6AADF0A762E3

Astrolirus patricki sp. nov. LSID: urn:lsid:zoobank.org:act:0F238379-CD10-4E51-A74A-0FEF055D72A5.

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
