# Peer review of "A new sponge-associated starfish, Astrolirus patricki sp. nov. (Asteroidea: Brisingida: Brisingidae), from the northwestern Pacific seamounts"

_PeerJ, doi:10.7717/peerj.9071_

## Round 0.1 · original submission · Major Revisions

I have heard back from two reviewers, both of whom have provided constructive comments to help you improve your work. While I have chosen major revisions needed for your work, I think the comments should be relatively easy to respond to, and look forward to seeing your revised work.

·

Basic reporting

fine.

Experimental design

fine.

Validity of the findings

yes.

Additional comments

The MS is well-written and presents the data clearly and with an excellent description for a difficult group of animals.

My only real issue is with the Discussion’s treatment of A. patricki’s association with sponges. Other than their use as substrate, I don’t see any the basis of why sponges should be considered any more closely associated with the brisingid species in question than any other substrate. The author clearly finds this to be an important relationship, going so far as to name the species after a cartoon character which is friendly with sponges.

But even the author outlines how there seems to be no apparent relationship between the two.. there is no feeding or gain by a specific mutualism.

I agree with and think positively of the notion that brisingids could be taking advantage of affected water flow created by the sponge as substrate..but that is more association with the general physical environment rather than a specific biological relationship which closely links this species of brisingid with the sponge. Brisingids are often found on corals or rock faces in current flow. Was there a statistically or anecdotally higher incidence of this species on sponges?? This could be entirely an incidental occurrence.

It seems to me that there should be a stronger basis to imply such a hypothesis. It is plausible..but I believe wording for this notion needs to be scaled back. And while I have no issues with naming the species after “Patrick” it should be borne in mind that if there is no association, the species epithet is a bit misleading.

Reviewer 2 ·

Basic reporting

This study is about integrative molecular phylogeny and morphological taxonomy for a deep sea asteroid species with good English writing. The authors showed the existence of new species and this manuscript might be an interest to the readers of Peer J and the basic methodology of molecular phylogeny and taxonomic handlings are generally acceptable and support the conclusions of this manuscript. But I think the manuscript needs to be improved with more information and interpretation with concrete pieces of evidence in molecular analysis and morphological observations.

Experimental design

The molecular phylogenetic analysis based on five gene regions but only three genes were concatenated. Considering that the increased number of genes provides more accurate phylogeny, concatenated genes analysis for all five genes should be done.

The “Description” was presented as a morphological summary of all examined specimens. However, the taxonomic description basically should be based on the holotype and other “variation” sections should be established in the manuscript. The description of the holotype is important because it is the name-bearing specimen. Photos and illustrations well help readers to understand the morphology of animals in taxonomic descriptions. But in the present figures (especially in Figs 2C-E; 3D, E), no closed up photos which enable us to understand the tiny morphology of ossicles are provided and it prevents readers, especially non-asteoird specialists, from smooth understanding the asteroid’s morphological features.

Thus I recommend authors to analyze molecular phylogeny based on multiple genes and improve the description part.

Validity of the findings

The “Habitat association with sponges” seems to me unnecessarily. The concrete data is a personal data of examination of stomach contents of a paratype (L248, 249). Other parts are the only review of the literature. If this part is not provided in this manuscript, validation of the description of the present new species is unquestioned. I recommend omitting this part unless you discuss this with your more concrete data (for example your phylogenetic trees).

Additional comments

Please see the other comments on my PDF file.

Annotated reviews are not available for download in order to protect the identity of reviewers who chose to remain anonymous.

---

## Round 0.2 · accepted · Accept

The manuscript has been well edited, and asides from one small comment from the reviewer, ready to be published. Thank you for your hard work.

Reviewer 2 ·

Basic reporting

My suggestions are all corrected and the manuscript is now improved.

Experimental design

Increasing molecular markers in the phylogenetic tree and additions of more detailed figures make clear the confidences of new species. No more problems have been found in the experimental design in this manuscript.

Validity of the findings

Discovery of such a very rare deep-sea new species is important to understand the deep-sea ecosystem. Has a good scientific impact.

Additional comments

My only minor request is as follow:

L151, "Description" should be "Description of the holotype"